# Cellular Uptake and Cytotoxicity of *Clostridium perfringens* Iota-Toxin

**DOI:** 10.3390/toxins15120695

**Published:** 2023-12-11

**Authors:** Masahiro Nagahama, Masaya Takehara, Soshi Seike, Yoshihiko Sakaguchi

**Affiliations:** 1Department of Microbiology, Faculty of Pharmaceutical Sciences, Tokushima Bunri University, Yamashiro-cho, Tokushima 770-8514, Japan; mtakehara@ph.bunri-u.ac.jp (M.T.); ysakaguchi@ph.bunri-u.ac.jp (Y.S.); 2Laboratory of Molecular Microbiological Science, Faculty of Pharmaceutical Sciences, Hiroshima International University, Kure, Hiroshima 737-0112, Japan; s-seike@hirokoku-u.ac.jp

**Keywords:** *Clostridium perfringens* iota-toxin, internalization, membrane repair, endocytosis

## Abstract

*Clostridium perfringens* iota-toxin is composed of two separate proteins: a binding protein (Ib) that recognizes a host cell receptor and promotes the cellular uptake of a catalytic protein and (Ia) possessing ADP-ribosyltransferase activity that induces actin cytoskeleton disorganization. Ib exhibits the overall structure of bacterial pore-forming toxins (PFTs). Lipolysis-stimulated lipoprotein receptor (LSR) is defined as a host cell receptor for Ib. The binding of Ib to LSR causes an oligomer formation of Ib in lipid rafts of plasma membranes, mediating the entry of Ia into the cytoplasm. Ia induces actin cytoskeleton disruption via the ADP-ribosylation of G-actin and causes cell rounding and death. The binding protein alone disrupts the cell membrane and induces cytotoxicity in sensitive cells. Host cells permeabilized by the pore formation of Ib are repaired by a Ca^2+^-dependent plasma repair pathway. This review shows that the cellular uptake of iota-toxin utilizes a pathway of plasma membrane repair and that Ib alone induces cytotoxicity.

## 1. Introduction

The clostridial binary actin-ADP-ribosylating toxins, encompassing *Clostridium perfringens* (Ia and Ib), *Clostridium botulinum* (C2I and C2II)*, Clostridium spiroforme* (Sa and Sb) and *Clostridioides difficile* (CDTa and CDTb), are composed of two independent proteins: the catalytic protein and the binding protein. The binding proteins specifically recognize different membrane receptors and are implicated in the entry of catalytic proteins into the cytosol [1,2,3,4,5]. The catalytic protein mono-ADP-ribosylates globular G-actin at arginine-177. Clostridial binary toxins induce the actin cytoskeletal disassembly and round-up of intoxicated cells. In addition, it has been reported that the binding protein alone exhibits cytotoxic effects [1,2,3,4,5]. The clostridial binary toxins are associated with the vegetative insecticidal proteins (*Bacillus thuringenesis* and related species) and anthrax toxins (*Bacillus anthracis*) [1,2]. *Bacillus* and *Clostridium* binary toxins show some relatedness; the binding components possess similar structures and action mechanisms [2]. They utilize the same mechanism to introduce the catalytic proteins into the cytoplasm mediated by the pore-forming activity of the binding proteins through the endosomal membranes. This review presents the progress made in recent years toward understanding the cellular internalization of iota-toxin and investigates the mechanism of cytotoxicity of Ib.

Iota-toxin is almost exclusively produced by *C. perfringens* type E strains. Infections with type E strain can cause antibiotic-associated enterotoxemia in rabbits and hemorrhagic enteritis and sudden death in calves and lambs [6,7,8]. Intestinal bacterial overgrowth of *C. perfringens* type E in bovines under certain conditions (overfeeding, stress, cold etc.) produces iota-toxin in the intestinal tract, which then attacks the intestinal mucosa facilitating intestinal injury such as a change in the epithelial permeability and cell necrosis. Iota-toxin consists of two non-linked protein components: Ia, the enzymatic ADP-ribosyltransferase which modifies G-actin, and Ib, which binds to the cell membrane and promotes the entry of Ia into the host cell cytoplasm [1,2,3,4,5]. Ib is produced as an inactive precursor (100 kDa) [1,2,3,4,5]. The active form of Ib (80 kDa) is produced by proteolytic cleavage with trypsin or chymotrypsin, resulting in the removal of a 20 kDa N-terminal region from the inactive precursor [1,2,3,4,5]. These proteases also cleavage short peptides (9 to 13 residues) from the N-terminal of the Ia precursor to generate an active form of Ia. Moreover, lambda protease produced by certain *C. perfringens*-type E strains activates iota-toxin [1,2].

Ia mono-ADP-ribosylates non-muscle β/γ-actin and skeletal muscle α-actin. Ia promotes cleavage of the *N*-glycosidic linkage between the nicotinamide group and *N*-ribose group of nicotinamide adenine dinucleotide (NAD^+^-glycosyl hydrolase) and transfer of the ADP-ribose group to G-actin (ADP-ribosyltransferase). Ia possesses three conserved motifs: the aromatic residue-Arg motif located in the catalytic center, the Glu-X-Glu motif, and the hydrophobic residue-Ser-Thr-Ser motif. The Glu-X-Glu motif, located in the ADP-ribosylating turn–turn loop, is crucial for the catalytic activity [9]. Results from site-specific mutagenesis studies of Ia showed that Glu-378 within Glu^378^-X-Glu^380^ is crucial for NAD^+^-glycosyl hydrolase and that Glu-380 within Glu^378^-X-Glu^380^ and Arg-295 within aromatic residue-Arg^295^ play important roles in the catalytic reaction of ADP-ribosyltransferase [9]. Structurally, Ia is segregated into two domains, one with the N-domain (1 to 210 residues) involving interaction with Ib and a second with the C-domain (211 to 413 residues) responsible for the enzymatic activity of mono-ADP-ribosyltransferase specific for actin [9]. On the basis of the crystal structure of Ia in a Michaelis complex with actin and a nonhydrolyzable NAD^+^ analogue, 62-Tyr on loop I and 248-Arg on loop II in Ia directly participate in the actin–Ia interface. Based on the structure of the Ia–actin complex, 378-Glu on the Glu-X-Glu motif of Ia is located in close proximity to 177-Arg in actin, and the ADP-ribosylation of 177-Arg in actin progresses by an SN1 reaction through an initial oxocarbenium ion intermediary body and a subsequent cationic intermediary body by alleviating the strained form of the initial oxocarbenium ion [9,10]. Yamada et al. reported the cryo-EM structures of the translocation channel Ib oligomer and Ia–Ib complex [11]. One Ia molecule is associated with one Ib oligomer. The N-terminal domain of Ia interacts with the Ib oligomer through several sites [11].

Ib shows meaningful homology to C2II of the C2 toxin (39.0% similarity overall) and the protective antigen of anthrax toxin (54.5% similarity overall) [12]. Ib, C2II and the protective antigen of anthrax toxin bind equally to host cell receptors on the cell surface and associate with the enzymatic components, transporting their internalization into host cells [1,3]. Therefore, they share a similar architectural organization and a mechanism of action. The structures of Ib and its homologues exhibit four distinct domains. Each domain of Ib has different functions, such as Ia binding (domain 1), oligomerization (domain 2), pore formation (domain 3), and binding to the receptor (domain 4). Domains 2 and 3 play roles in the pore formation of Ib [1,2,3,4,5]. Domain 1 in the amino-terminal part of Ib includes the binding site for Ia. The conserved Ca^2+^-binding motif present in domain 1 of Ib is responsible for the binding of Ib with Ia in the presence of Ca^2+^ [9]. Domain 4 (residues 421-664 of Ib, Ib421-664) blocks Ib binding to the host cells [5,13] and is predominantly localized to lipid raft domains. Ib421-664 binds to the lipolysis-stimulated lipoprotein receptor (LSR) without causing cytotoxic activity [14,15,16]. It appears that because Ib421-664 binds to LSR which is localized throughout the plasma membrane, LSR, which is associated with Ib, migrates into lipid raft domains of the plasma membrane. LSR is reconfirmed as angulin-1, the decisive factor for the membrane localization of tricellulin at tricellular tight junctions [17,18,19], and Ib421-664 is identified as the tricellular tight junction-specific modifier [20,21,22,23]. Ib421-664 also reduced epithelial barrier function and the expression level of LSR, and it promoted cell penetration in a pancreatic tumor cell line (HPAC) [21]. Ib421-664 reversibly modulates the epithelial barrier and cell penetration at tricellular contacts via JNK/cofilin/actin cytoskeleton dynamics [22]. Ib421-664 is used for drug delivery applications [20,23]. Ib421-664 binds to LSR, which promotes the partial removal of tricellulin from tricellular tight junctions. Since tricellulin is known to be a sealing tight junction protein [19] whose existence is a modification factor for the macromolecular diffusion barrier function of the tricellular tight junctions [20,21,22,23], the removal of tricellulin due to the disappearance of LSR by Ib421-664 from a tricellular tight junction enhances macromolecular transport through the junction. Treatment with Ib421-664 alters the local localization of tricellulin, opening the tricellular tight junction. Ib421-664 represents an absorption enhancer targeting the tricellular tight junction. In contrast, LSR-expressing brain microvascular endothelial cells exist in the blood–brain barrier (BBB) [23]. The intravenous injection of Ib421-664 enhances the vascular permeability of the blood–brain barrier and allows the transient transport of subsequently administered antisense oligonucleotides into the mouse central nervous system, promoting the silencing of target RNA. Delivery via the tricellular tight junction in the blood–brain barrier by Ib421-664 offers a new antisense oligonucleotide delivery system for neurological disorder therapies [23].

## 2. Binding and Internalization of Iota-Toxin

Iota-toxin internalizes into endosomes of sensitive cells and causes cell death by employing the endocytic membrane trafficking pathways [5] (Figure 1). LSR is a cellular receptor for Ib [11,15]. LSR is comprised of a surface-exposed N-terminal Ig-like domain followed by a single-pass transmembrane region and a C-terminal cytosolic tail region [17,18,19]. LSR is first found in hepatic cells, where it mediates the clearance of triglyceride-rich lipoprotein and low-density lipoproteins [17,18]. The siRNA-mediated knockdown of LSR in iota-toxin-sensitive cells leads to a decreased binding of Ib and iota toxin-caused cytotoxic effect, which confirms the specific binding of Ib to LSR [16]. Moreover, Ib induces the accumulation of LSR in lipid raft domains. We previously reported that the N-terminal (amino acid 1–14) deletion mutant of LSR exhibits a marked reduction in the binding of Ib [16]. Based on these findings, Ib can interact with 15 N-terminal extracellular amino acids of LSR [16]. The C-terminal domain of Ib (Ib421–664) is concerned with LSR-binding [5,20,21,22,23]. Ib associates with LSR on the cell surface via the C-terminal region of Ib and clusters in plasma membrane lipid rafts, and then Ia associates with the Ib oligomer formed on lipid rafts to enter host cells by receptor-mediated endocytosis [5,14,16]. The Ib-LSR complex is internalized and colocalized in endocytic vesicles. It is known that LSR is responsible for the binding of apolipoproteins, promoting their entry. Ib internalizes the target cells via the route of LSR-dependent endocytosis. In contrast, it has been reported that Ib internalizes the target cells through cell-surface antigen CD44-dependent endocytosis [24]. Moreover, CD44 is predominantly found in lipid raft microdomains obtained from target cells incubated with Ib [25], and CD44 facilitates the accumulation of LSR into lipid raft microdomains [26].

Host cells efficiently overcome pore-forming toxin (PFT) attacks by a membrane repair mechanism to eliminate pores [27,28,29,30]. The repair mechanism is evoked by Ca^2+^ entry through the transmembrane pores formed by PFT. Namely, PFTs take advantage of the plasma membrane repair mechanisms to gain access to the host cell cytosol. This event triggers the exocytosis of lysosomes and the extracellular secretion of the lysosomal acid sphingomyelinase (ASM) and lysosomal protease cathepsins. PFT-induced host cell membrane perforation causes an influx of extracellular Ca^2+^ [31,32]. The Ca^2+^-mediated exocytosis of lysosomes accelerates lesion elimination by endocytosis. The hydrolytic cleavage of sphingomyelin catalyzed by ASM on the outer leaflet of the cell membrane leads to the formation of a ceramide-rich domain in the cell membrane [31,32]. This ceramide-rich domain generates membrane invagination and promotes a typical form of endocytosis, which internalizes the toxin pores and leads to cell membrane repair. Furthermore, lysosomal protease cathepsins B and L secreted into the extracellular space have the capacity to degrade numerous extracellular matrix proteins in plasma membranes [33]. This removal of plasma membrane-associated proteins by cathepsins plays an important role in the rapid access of ASM to the plasma membrane sphingomyelin. Ib is immediately internalized under situations of Ca^2+^-containing medium but not Ca^2+^-free medium. We reported the increase in intracellular Ca^2+^ entry by Ib exposure at an initial stage of the Ib intoxication process [34] (Figure 1). The influx of extracellular Ca^2+^ through oligomeric Ib pores triggers rapid lysosomal exocytosis, exposing the lysosome-derived membrane proteins on the outer leaflet of the plasma membrane and releasing lysosomal enzymes [34]. Ib causes the release of ASM derived from lysosomes into the extracellular space. Consequentially, Ib gives rise to ceramide production. The cytotoxicity of iota-toxin against sensitive cells is blocked by ASM blockers and ASM knockdown by siRNA. Ib also promotes the extracellular release of lysosomal cysteine protease cathepsin B and L [35]. Cysteine protease blocker E64 and siRNA knockdowns of cathepsin B and L reduce the cytotoxicity of iota-toxin. Cathepsins B and L are responsible for the proteolytic cleavage of the host extracellular matrix proteins in cytoplasmic membranes [33]. The elimination of extracellular membrane proteins allows access by ASM to sphingomyelin in the plasma membrane, as mentioned above. Likewise, Ia becomes more accessible to the Ib oligomer on the plasma membrane. In contrast, it has been reported that cathepsins released from lysosomal exocytosis are responsible for the activation of extracellularly released ASM [33]. ASM released by Ib was activated by cathepsins B and L coming from lysosomal exocytosis, indicating that the release of lysosomal cathepsins B and L caused by Ib is related to the activation of ASM and accessibility of ASM to the cytoplasmic membranes. From these findings, cathepsin B and L play roles in the internalization of iota-toxin into target cells [35]. That is, when Ib forms transmembrane pores in target cells, the Ca^2+^-dependent membrane repair is induced, accelerating the endocytosis of iota-toxin. The iota-toxin is internalized in cells through an innate defense response, protecting the target cells against PFT. Ib takes advantage of the membrane repair system as an internalization process. Ib is taken up into sensitive cells via the Ca^2+^-mediated lysosomal exocytosis of ASM and cathepsins. However, Ib does not cause any elevation of neutral sphingomyelinase (NSM) activity [35]. Furthermore, this iota-toxin-induced cytotoxic effect is not inhibited by an NSM blocker. ASM-mediated ceramide generation induces a negative membrane curvature in the outer layer. This curvature promotes an inward movement of endocytic vesicles [36]. ASM plays a critical role in the cellular internalization of iota-toxin. In contrast, NSM participates in the production of ceramide-enriched domains in the inner layer of the cytoplasmic membrane so that the membrane results in the formation of an outward curvature that is involved in the outer-layer shedding of injury [36]. The incubation of MDCK cells with Ib does not cause cytoplasmic membrane blebbing leading to the shedding of membrane-bound Ib. NSM is not related to the action of iota-toxin [35].

*C. botulinum* C2 toxin comprises actin ADP-ribosyltransferase (C2I) and C2II binding components [1,2,3,4]. Activated C2II (C2IIa) binds to asparagine-linked carbohydrate receptors on target cells and forms oligomers in membrane rafts [1,2,3,4]. The docking of the C2I oligomer to the C2IIa oligomer on lipid rafts causes the activation of the phosphatidylinositol 3-kinase (PI3K)-Akt signaling pathway, which then facilitates the entry of C2 toxin into target cells via lipid raft microdomains [5]. The C2 toxin can readily enter target cells in the presence of Ca^2+^ but not in the absence of Ca^2+^ [37]. The cytotoxicity of the C2 toxin was augmented in Ca^2+^-containing medium. C2IIa induces Ca^2^ influx through C2IIa pores from extracellular medium to target cells [37]. C2IIa-induced Ca^2^ influx into cytoplasm then promotes lysosomal exocytosis, provoking the release of ASM and cathepsins to the extracellular medium as well as Ib [37,38]. Cysteine protease inhibitor E64 inhibits C2 toxin-induced cytotoxicity and C2IIa-induced ASM activity. The knockdown of lysosomal cysteine protease cathepsin B by siRNA inhibits the cytotoxicity of C2 toxin [38]. From these findings, the C2 toxin induces both cathepsin B and ASM release via lysosomal exocytosis. Cathepsin B proteolytically activates ASM and promotes the accessibility of ASM and C2I to cytoplasmic membranes via cell surface protein cleavage by cathepsin B [38]. ASM generates ceramide-rich domains. The domains invaginate the host cell membrane, constructing endosomes that promote C2 toxin entry into the cytosol [37].

## 3. Cellular Trafficking of Iota-Toxin

The target cell entry of iota-toxin utilizes a process of membrane repair pathway that includes the Ca^2+^-dependent exocytosis of lysosomes, the delivery of cathepsins and ASM to the outer leaflet of the plasma membrane, and a quick form of endocytosis that internalizes LSR-binding toxin [27,28,29,30]. Ib recognizes LSR in the plasma membrane of target cells, oligomerizes into heptameric structures on lipid raft domains, and docks with Ia [5,39]. After endocytosis through membrane repair, a clathrin-independent and Rho-dependent pathway [40,41], the toxin (Ia–Ib complex) organizes the host endosomal trafficking pathway [5,34,35]. An internalized iota-toxin is initially delivered to early endosomes [5]. Some Ib is transported to Rab11-positive recycling endosomes and sent to the cytoplasmic membrane. This recycling process is crucial for Ib to expand the effective Ia incorporation into target cells. A large quantity of Ib is translocated from early endosomes to late endosomes and is delivered to lysosomes for the degradation of Ib [5] (Figure 1). Lysosomes are related to the breakdown of a large number of extracellular proteins that enter cells through endocytosis [42]. Accordingly, by means of fusion of endosomes including Ib with lysosomes, the luminal region of Ib is destroyed via the lysosomal proteolysis but not the transmembrane region of Ib. Lysosomes migrate from cytosol to cytoplasmic membranes through a Ca^2+^-mediated pathway and fuse with the cytoplasmic membranes, showing that lysosomes are the Ca^2+^-dependent exocytotic compartments [42,43]. As Ib causes Ca^2+^ entry into target cells during endocytosis, an increase in the intracellular Ca^2+^ level by Ib promotes the fusion of lysosomes with cytoplasmic membranes, and digested Ib is trafficked to cytoplasmic membranes.

Ia transits through Ib pores into the cytosol [1,2,3,4]. The mode of action of Ia transport across the membrane of early endosomes concerns the passage of unfolding Ia via the pore-forming oligomeric Ib within an acidic environment. Bafilomycin A1, a blocker of the vacuolar-type H^+^-ATPase and therefore of endosomal acidification, decreases the passage of Ia into the cytoplasm and subsequent cytotoxic effects [5]. Based on Ia-Ib oligomer structural analysis, Yamada et al. reported that the binding of the Ia oligomer to the Ib oligomer causes Ia N-terminal α-helix tilting and partial unfolding, which is related to Ia translocation through Ib pores, indicating that the Ib oligomer (Ib pore) contributes as an unfolding chaperone for Ia [11]. The Ib pore provides a new mechanism for the unfolding of the N-terminal region of Ia. The acidification of endosomes promotes the entry of an unfolded N-terminal region of Ia into an Ib pore. This N-terminal unfolding of Ia induced by the Ib oligomer is important for Ia translocation across the endosomal membrane to the cytosol through Ib pores [11] (Figure 1). The transmembrane transport or cytoplasmic refolding of unfolded Ia is promoted by host cell factors including the molecular chaperones heat shock protein 90 (Hsp90) and Hsp70 as well as the peptidyl-prolyl *cis/trans*-isomerase (PPIase) of cyclophilin A (CypA) and FK-506 binding protein (FKBP) [44,45,46,47,48]. Inhibition of the activities of Hsp90, Hsp70, CypA and FKBP using specific pharmacological blockers results in the attenuated cytotoxicity of iota-toxin. Treatment with mixtures of four specific blockers markedly delayed the intoxication of cells with iota-toxin compared with the single blocker as well as mixtures of two or three blockers [48]. Hsp90, Hsp70, CypA and FKBP promote the translocation and protein folding by binding to unfolded Ia [44,45,46,47,48]. In the cytosol, Ia covalently transfers an ADP-ribose moiety from substrate NAD^+^ onto monomeric actin (G-actin) at argine-177 [1,2,3,4,5]. ADP-ribosylated actin generated by Ia behaves similarly to a capping protein [49]. The binding of ADP-ribosylated G-actin to the growing end (or barbed end) of filamentous actin inhibits the binding of additional G-actin due to steric hinderance by a covalently linked ADP-ribose moiety in ADP-ribosylated G-actin [49]. At the distinct terminal region of the filamentous actin, the liberation of G-actin is still underway. This ultimately leads to the overall depolymerization of the actin filaments and accelerates the rounding of target cells [49].

C2 toxin internalizes the target cells and causes cytotoxicity by employing endocytic membrane trafficking [12,50]. C2IIa binds an asparagine-linked carbohydrate receptor on the plasma membrane of target cells and forms a heptameric oligomer that binds C2I [51]. The C2I–C2IIa complex is internalized by receptor-mediated endocytosis via a lipid raft domain using the membrane repair pathway and transports to early endosomes [37,38,52,53]. The C2 toxin is also transported to early endosomes, where the entry of C2I into the cytoplasm occurs [5]. Acidification within the early endosomes triggers the cytoplasmic translocation of C2I. The entry of C2I is facilitated by Hsp90, Hsp70, CypA and FKBP as well as the translocation of Ia [54,55]. The combination of specific pharmacological blockers exerts a profound protective effect against the C2 toxin compared to iota-toxin [47,48], indicating that slight differences exist in the cytoplasmic translocation process between Ia and C2I. C2I then catalyzes the ADP-ribosylation of G-actin in the cytoplasm [1,2,3,4,5]. Some C2IIa is trafficked from the early endosome to Rab11-posotive endosomes, demonstrating that C2IIa is transported to recycling endosomes. The incubation of C2I to cells pretreated with C2IIa induces cell rounding [5]. C2IIa in recycling endosomes is sent to the cytoplasmic membrane, where it becomes capable of rebinding to C2I. The efficient recycling of CIIa from recycling endosomes is crucial for expanding the internalization of C2I. Some other C2IIa is transported from early endosomes to late endosomes and lysosomes for lysosomal degradation [5]. 

## 4. Cytotoxic Effect of Ib

It has been reported that Ib alone does not cause cell death [9,56]. In contrast, we reported that Ib induces the release of K^+^ from Vero cells, indicating that Ib binds to target cells and forms heptameric oligomers to generate K^+^-permeable ion channels and that the Ib oligomer invades the plasma membrane [5]. Ib induces carboxyfluorescein release from phosphatidylcholine-cholesterol liposomes and forms a heptameric oligomer in the liposomes, showing that Ib oligomerization and functional pore formation take place in lipid bilayers [5]. Furthermore, Ib induces a reduction in trans-epithelial electrical resistance (TEER) in polarized Caco-2 cells. Ib causes an efflux of K^+^ and the entry of Na^+^ in Caco-2 cells, demonstrating that the Ib oligomer changes the plasma membrane permeability, allowing the passage of K^+^ and Na^+^ ions [57]. We showed that Ib alone is cytotoxic to human epithelial carcinoma A431 and human lung adenocarcinoma A549 cells [58]. Ib induces cell swelling, the dysfunction of mitochondria, ATP depletion, elevated propidium iodide entry and a reduction in survival rate. Bax and Bak proteins belong to the Bcl-2 family and are the main modulators of intrinsic apoptosis. Although Ib induces the activation of Bax and Bad and the release of cytochrome C leading to apoptotic cell death, Ib does not activate caspase-3, and the broad-spectrum caspase inhibitor z-VAD-fmk does not inhibit Ib-induced cytotoxicity. From these results, Ib causes cell necrosis but not apoptosis [58]. We reported that Ib binds to LSR in cell membranes of MDCK cells and then moves to lipid raft microdomains [16] and that the Ib oligomer formed in the rafts enters MDCK cells via endocytosis [16,58]. In contrast, Ib forms oligomers on the non-raft membrane fraction of iota-toxin-sensitive A431 cells [58]. In A431 cells, Ib is located on the cell surface during the intoxication process and is not internalized into cytoplasm [58]. Namely, the ability of mammalian cells to survive cell membrane perforation by an Ib oligomer is likely dependent upon the cellular function to internalize Ib. The internalization of Ib is required for host cell survival and indicates a role for endocytosis as an innate cellular defense response toward membrane perforations by PFT [31,32].

*Clostridioides difficile* binary toxin (CDT) is composed of an enzyme component (CDTa) and a binding component (CDTb) [59]. CDTb forms an ion channel in a black lipid bilayer [60,61]. CDTb binds to the cell surface of LSR, promoting the docking and entry of CDTa via receptor-mediated endocytosis [26,59]. CDTb alone also exhibits cytotoxic activity [61]. CDTb induces the cell death of Caco-2 cells [62]. CDTb-mediated cytotoxicity is dependent on the presence of LSR [61,62]. CDTb causes the clustering and accumulation of LSR into lipid rafts [26]. CDTb accumulation facilitates the oligomerization of CDTb. CDTb pores in the plasma membrane serve to induce cytotoxic effects [61]. C2IIa of the *C. botulinum* C2 toxin forms cation-selective channels in artificial phospholipid bilayer membranes [3]. Chloroquine and its analogs (aminoquinolinium salts) can inhibit ion channels created in lipid bilayer membranes by C2IIa [63]. C2IIa induces the extracellular efflux of radioactive rubidium-86 ions, a radioactive analogue of K^+^, from living cells, indicating that C2IIa forms pores in cytoplasmic membranes [52,64,65]. CIIa alone induces morphological alterations and cytotoxicity in primary human polymorphonuclear leukocytes (PMNs) [66]. C2IIa rapidly causes Ca^2+^ influx in PMNs, indicating that the pore formation of the C2IIa oligomer in cell membranes is important for membrane disruption and reduces PMNs chemotaxis [66]. On the basis of these investigations, the binding components of clostridial binary actin-ADP-ribosylating toxins induce cytotoxicity by pore formation in the cytoplasmic membranes of host cells.

## 5. Effect of Iota-Toxin on Animals

Iota-toxin demonstrates lethal and dermonecrotic activity in animals [1,2,3,4,9]. These activities of iota-toxin are caused by the cooperative effect between Ia and Ib [1,2,3,4,9]. From extensive site-specific mutagenesis studies of Ia, Ia mutants lacking ADP-ribosyltransferase activity do not cause lethal activity in the presence of Ib in mice [9]. Therefore, it seems that the lethality of iota-toxin is closely related to the ADP-ribosyltransferase activity of Ia. The effect of separate administrations of Ia and Ib on the biological activities of iota-toxin was previously reported [67]. The lethal activity of the iota-toxin in mice was seen after the intravenous administration of one component within 2 h of administration of the other. The intravenous injection of Ib to mice injected with rabbit polyclonal antiserum against Ia within 120 min after the intravenous administration of Ia did not lead to death [67]; i.e., the Ia antiserum recognized Ia in vivo within 2 h after intravenous administration of Ia, indicating that Ia is present in blood flow. The injection of Ia to mice injected with rabbit polyclonal antiserum against Ib within 5 min after the intravenous administration of Ib caused death, demonstrating that antiserum against Ib cannot neutralize Ib under these conditions. It seems that after binding to its receptor, Ib progresses toward a step which is not blocked by the antiserum, showing that the Ia–Ib complex is internalized to the target organ [67]. When Ia and Ib were intradermally administered at distinct skin sites in adult guinea pigs, dermonecrotic activity was recognized at the area of administration of Ib but not Ia. Moreover, intraperitoneal administrations of Ib and Ia after the intradermal administration of Ia and Ib, respectively, induced dermonecrotic activity at the area of intradermal administration of Ib but not that of Ia. Therefore, Ia is present in the blood flow, and Ib binds to its receptor in the administered skin area, indicating that the migration of Ia to the Ib-administered area is essential to the dermonecrotic activity caused by separate administrations of the two components [67]. This demonstrates that Ia associates with Ib bound to target organs, leading to lethality and dermonecrosis.

## 6. Conclusions

Iota-toxin belongs to the clostridial binary actin-ADP-ribosylating toxin family. The toxin is able to enter cells and transport an ADP-ribosylating component into the cytosol. Iota-toxin binds to a host cell receptor LSR. Pores formed by oligomers of Ib in lipid rafts facilitate Ca^2+^ influx and lysosomal exocytosis. Iota-toxin enters the host cell via the plasma membrane repair mechanism. Ib alone exhibits a cytotoxic effect by pore formation in the plasma membrane of host cells. Blockers of binding, internalization and intracellular transport of iota-toxin are potential therapeutic drugs for the treatment of infectious diseases.

## Figures and Tables

**Figure 1 toxins-15-00695-f001:**
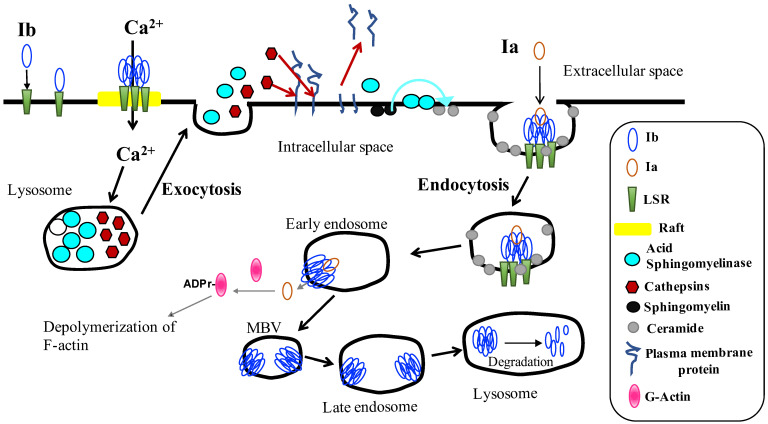
**Cellular uptake of iota-toxin.** Ib binds with LSR (lipolysis-stimulated lipoprotein receptor), moves to lipid rafts and forms an Ib oligomer (pore). Ca^2+^ influx through Ib pores quickly promotes lysosomal exocytosis and the release of cathepsins and acid sphingomyelinase into the extracellular space. Cathepsins proteolytically degrade the cell surface proteins and facilitate the access of ASM or Ia to the membrane surface. ASM cleaves sphingomyelin and generates ceramide in the outer leaflet of the plasma membrane, which promotes endocytosis. The Ia and Ib complex internalizes to the target cells. Iota-toxin is transported to the early endosome, where acidification accelerates the release of Ia into cytoplasm. Ia ADP-ribosylates G-actin in the cytoplasm, causing cytotoxicity. Ib is trafficked to late endosomes. Ib is transported to lysosomes for degradation.

## Data Availability

Not applicable.

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
