# Peer review of "Cellular Uptake and Cytotoxicity of *Clostridium perfringens* Iota-Toxin"

_toxins, 2023, doi:10.3390/toxins15120695_

Round 1

Reviewer 1 Report

Comments and Suggestions for Authors

This paper is a review about the cellular uptake and cytotoxicity of Clostridium perfringens iota-toxin. This paper is clear and provides an overview of the binary iota toxin as the proteins, on one hand, one with catalytic activity (Ia), and on the other, one (Ib) with binding to its receptor (LSR), present in lipid rafts of the host cell. In this article, protein Ib is introduced as a pore-forming toxin (PFT) that imparts toxicity to the cell on its own. Simultaneously, the cell defends itself by undergoing endocytosis of the protein bound (Ib) to its plasma membrane with the aim of repairing the membrane.

As a general comment, in terms of the paper's format, it is generally written with little fluency and composed of the combination of sentences separated by commas or fullstops (I miss connectors between sentences that provide fluidity to the text). Perhaps in some paragraphs, it could be rewritten to provide more ease for the reader. It is worth noting that this is just one perspective and that everyone has their own style.

There are minor aspects and comments to take into account.

-       Line 31, please, include a cite after (Bacillus anthracis), where the association with insecticidal proteins and anthrax toxin is explicitly stated.

-       Line 38, please, remove the begginig of the sentence (of rabbits), it is redundant because it is understood later on.

-       Line 50, please, include a cite at the end of the sentence (iota-toxin).

-       Line 77, please, include a cite after (into host cells).

-       Line 96, please, include a cite after (delivery applications).

-       Line 97, please, include a cite after (tight junctions).

-       Paragraphs from line 103 to 109, if I have undrestood well, injection of Ib421-664 enhances de vascular permeability of the BBB, it means that it produces a damage of the the BBB or an alteration of its structure? It would produce an acute edema in the brain? In that case, the use of Ib421-664 would be safe in vivo? Please, comment on that.

-       First paragraph of point 2 (lines 111 to 119). It is said that LSR is the receptor for iota-toxin, and siRNA-mediated knockdown for LSR marked a reduction of the Ib binding to the plasma membrane (but not an abolishment). Do you know if there are another experiments to confirm that? For exemple, Crispr-Cas9 for LSR, preincubation with LSR antibodies prior to Ib incubation or the use specific LSR blockers.

-       Line 125, is the oligomer formation a pore, or a pre-pore? My question is: Ia enters into the host cell associated with Ib by the endocytic way, and at the same time, Ib has been a pore inserted in the plasma mebrane producing a deregulation of ions, prior to internalization? Or Ib is only a pre-pore which is inserted and produces a pore once in the endosome membrane after acidification?

-       Line 139, after the influx of the extracellular calcium there is a formation of endosomes in order to remove the Ib from the plasma membrane. There are another PFT toxins (as the alptha toxin produced by Staphylococcus aureus, or the perfringolysin from Clostridium perfringens) which it is known that the host cells infected by these toxins produce the secretions of extracellular vesicles (blebbing) in order to release the pores from the plasma membrane, quite similar to iota-toxin, why not iota-toxin could do the same? Or may be both, an endocytic and an exocytic pathway? Can you comment on that, please?

-       Line 177, please, include a cite after (endocytic vesicles).

-       Line 260, which are the specific pharmacological blockers? Blockers for Hsp90, Hsp70 and CypA? I have not understood. It this case, may be the difference between C2 toxin and iota-toxin would be that C2 gets into the cell via endocytic pathway exclusivelly, and iota-toxin, may be not, or partially not, and the extracellular pathway would come into play. What do you think?

-       Line 297, please, include a cite after (PFT).

-       Line 305, please, include a cite  after (cytotoxic effects).

-       Line 306, please, include a cite after (phospholipid bilayer membranes).

Author Response

This paper is a review about the cellular uptake and cytotoxicity of Clostridium perfringens iota-toxin. This paper is clear and provides an overview of the binary iota toxin as the proteins, on one hand, one with catalytic activity (Ia), and on the other, one (Ib) with binding to its receptor (LSR), present in lipid rafts of the host cell. In this article, protein Ib is introduced as a pore-forming toxin (PFT) that imparts toxicity to the cell on its own. Simultaneously, the cell defends itself by undergoing endocytosis of the protein bound (Ib) to its plasma membrane with the aim of repairing the membrane.

As a general comment, in terms of the paper's format, it is generally written with little fluency and composed of the combination of sentences separated by commas or fullstops (I miss connectors between sentences that provide fluidity to the text). Perhaps in some paragraphs, it could be rewritten to provide more ease for the reader. It is worth noting that this is just one perspective and that everyone has their own style.

Answer: Thank you for your recommendation.

There are minor aspects and comments to take into account.

-       Line 31, please, include a cite after (Bacillus anthracis), where the association with insecticidal proteins and anthrax toxin is explicitly stated.

Answer: OK. We agree with you. We added references [1,2] in line 37, as pointed out.

-       Line 38, please, remove the begginig of the sentence (of rabbits), it is redundant because it is understood later on.

Answer: OK. We agree with you. We deleted the begginig of the sentence (of rabbits)in line 43, as pointed out.

-       Line 50, please, include a cite at the end of the sentence (iota-toxin).

Answer: OK. We agree with you. We added references [1,2] in line 56, as pointed out.

-       Line 77, please, include a cite after (into host cells).

Answer: OK. We agree with you. We added references [1,3] in line 83, as pointed out.

-       Line 96, please, include a cite after (delivery applications).

Answer: OK. We agree with you. We added references [20,23] in line 102, as pointed out.

-       Line 97, please, include a cite after (tight junctions).

Answer: OK. We agree with you. We added references [19] in line 104, as pointed out.

-       Paragraphs from line 103 to 109, if I have undrestood well, injection of Ib421-664 enhances de vascular permeability of the BBB, it means that it produces a damage of the the BBB or an alteration of its structure? It would produce an acute edema in the brain? In that case, the use of Ib421-664 would be safe in vivo? Please, comment on that.

Answer:  Previous reports have shown the following (Pharmaceutics 2020, 12, 1236, [Ref.16]). In mice, Ib421-664can reversibly reduce the integrity of the BBB in vivo. Mice treated with Ib421-664showed no abnormal behavior and showed normal liver and kidney functions according to histological and serum biochemical tests. Additionally, mice treated with Evans blue solution following Ib421-664showed no extravasation of Evans blue into the brain. Regarding drug delivery of b421-664, it is still in the study stage, but we think it will be further improved in the future.

-       First paragraph of point 2 (lines 111 to 119). It is said that LSR is the receptor for iota-toxin, and siRNA-mediated knockdown for LSR marked a reduction of the Ib binding to the plasma membrane (but not an abolishment). Do you know if there are another experiments to confirm that? For exemple, Crispr-Cas9 for LSR, preincubation with LSR antibodies prior to Ib incubation or the use specific LSR blockers.

Answer:  As for other LSR experiments, we don't know.

-       Line 125, is the oligomer formation a pore, or a pre-pore? My question is: Ia enters into the host cell associated with Ib by the endocytic way, and at the same time, Ib has been a pore inserted in the plasma mebrane producing a deregulation of ions, prior to internalization? Or Ib is only a pre-pore which is inserted and produces a pore once in the endosome membrane after acidification?

Answer:  Ib forms pore in cell membrane. Once boud to LSR, Ib assembles in heptamers which insert into the plasma membrane forming functional channels, allowing the movement of ions, and the translocation and endocytosis of Ia.

-       Line 139, after the influx of the extracellular calcium there is a formation of endosomes in order to remove the Ib from the plasma membrane. There are another PFT toxins (as the alptha toxin produced by Staphylococcus aureus, or the perfringolysin from Clostridium perfringens) which it is known that the host cells infected by these toxins produce the secretions of extracellular vesicles (blebbing) in order to release the pores from the plasma membrane, quite similar to iota-toxin, why not iota-toxin could do the same? Or may be both, an endocytic and an exocytic pathway? Can you comment on that, please?

Answer:  Pore formation of PFT in cell membrane causes the host cell injury. There is secretions of extracellular vesicles (blebbing) to repair the damage to the cell membrane caused by PFT. In MDCK cells, Ib causes Ca2+-induced exocytosis of lysosomes and rapid endocytosis to repair membrane. On the other hand, when Ib was treated with A431cells, the cells showed marked swelling similar to bleb formation (Infect.Immun. 79, 4253 (2011)). We think that the actions of Ib differ depending on the cell type (for example, differences in the abundance of receptors and rafts in each cells).

  -       Line 177, please, include a cite after (endocytic vesicles).

Answer: OK. We agree with you. We added references [36] in line 183, as pointed out.

-       Line 260, which are the specific pharmacological blockers? Blockers for Hsp90, Hsp70 and CypA? I have not understood. It this case, may be the difference between C2 toxin and iota-toxin would be that C2 gets into the cell via endocytic pathway exclusivelly, and iota-toxin, may be not, or partially not, and the extracellular pathway would come into play. What do you think?

Answer:  Specific inhibitors have been reported in a previous paper (Naunyn Schmiedebergs Arch. Pharmacol. 2021, 394, 941(Ref.47),Front. Cell Infect. Microbiol. 2022, 12, 938015 (Ref.48)). Hsp90 inhibitors are radicicol and geldanamycin. Hsp70 inhibitors are VER-155008 and HA9. CypA inhibitors are cyclosporine A and VK112.The difference between Ia and C2I, which we mentioned in the paper where the reviewer pointed out, is that Ia or C2I translocate from the endosomes through Ib pore or C2IIa pore into the cytoplasm. We cannot explain the detailed differences at this time, but we believe that structural analysis of this process will reveal them in the future.

-       Line 297, please, include a cite after (PFT).

Answer: OK. We agree with you. We added references [31,32] in line 302, as pointed out.

-       Line 305, please, include a cite  after (cytotoxic effects).

Answer: OK. We agree with you. We added references [62] in line 310, as pointed out.

-       Line 306, please, include a cite after (phospholipid bilayer membranes).

Answer: OK. We agree with you. We added references [3] in line 311, as pointed out.

Reviewer 2 Report

Comments and Suggestions for Authors

This manuscript provides a detailed review of research on the mode of action of Clostridium perfringens iota toxin.  It should be an important reference for the status of our understanding of the toxin.  With relatively minor edits, the manuscript is appropriate for publication.

The focus of the manuscript is primarily based on extensive work with various cell cultures. However, the consideration of effects in animal models (section 5) is limited and does not include likely effects in humans. This is particularly limiting in relation to effects observed in cell cultures but for which there are no available cells lines.  For example, lines 272-275 the authors report that Ib disrupts K+ channel which suggests that they may have an impact on nerve transmission.  Another area that is not mentioned in the review is similar potential modes of action in member of the Bacillus cereus group where diarrheal diseases are associated with cathepsins B and L. 

General.  I found is somewhat surprising that there is no mention in the text about the genetics of iota toxin such as the likelihood of it could transfer to other Clostridium perfringens strains or potentially to other bacterial species.

Lines 112-113.  While the abbreviation LSR {lipolysis-stimulated lipoprotein receptor) is defined in the abstract. It is not defined in the text until this sentence.

Author Response

This manuscript provides a detailed review of research on the mode of action of Clostridium perfringens iota toxin.  It should be an important reference for the status of our understanding of the toxin.  With relatively minor edits, the manuscript is appropriate for publication.

Answer: Thank you for your recommendation.

The focus of the manuscript is primarily based on extensive work with various cell cultures. However, the consideration of effects in animal models (section 5) is limited and does not include likely effects in humans. This is particularly limiting in relation to effects observed in cell cultures but for which there are no available cells lines.  For example, lines 272-275 the authors report that Ib disrupts K+ channel which suggests that they may have an impact on nerve transmission.  Another area that is not mentioned in the review is similar potential modes of action in member of the Bacillus cereus group where diarrheal diseases are associated with cathepsins B and L. 

Answer:  We wrote this paper based on the results of examining the mechanism of action of toxins using various cells. To date, infections caused by iota-toxin-producing C. perfringens type E have only been reported in animals, and there have been no reports of infections caused by the bacteria in humans. For the above reasons, we do not mention human infectious diseases. In lines 1-3, we described that rather than the Ib destroying the K channel, K ion released from Ib pore formed on the cell membrane. There are no reports that Ib affects neurotransmission. C. perfringens iota-toxin is not related to toxins produced by B. cereus. Therefore, we have not written about Bacillus toxins.

General.  I found is somewhat surprising that there is no mention in the text about the genetics of iota toxin such as the likelihood of it could transfer to other Clostridium perfringens strains or potentially to other bacterial species.

Answer:  Thank you for good suggestion. We submitted this paper to a special issue on ADP-Ribosylation and Beyondin Toxins. In this paper, we mainly described the mechanism of action of iota-toxin, an ADP-ribosylating toxin. The genetics of the toxin is left for other reviews.

Lines 112-113.  While the abbreviation LSR {lipolysis-stimulated lipoprotein receptor) is defined in the abstract. It is not defined in the text until this sentence.

Answer: OK. We agree with you. We deleted the lipolysis-stimulated lipoprotein receptorin line 118, as pointed out.

Reviewer 3 Report

Comments and Suggestions for Authors

The review is current, is written in clear scientific language, the structure of the review is good, the review cites 69 publications, most of them from the last 10 years. I believe that this review can be published in its current form.

Author Response

The review is current, is written in clear scientific language, the structure of the review is good, the review cites 69 publications, most of them from the last 10 years. I believe that this review can be published in its current form.

Answer: Thank you for your recommendation.

Reviewer 4 Report

Comments and Suggestions for Authors

Well written and very thorough review of the subject. It will make a nice contribution to the field.

Comments on the Quality of English Language

Just a few minor syntax errors for correction.

Author Response

Well written and very thorough review of the subject. It will make a nice contribution to the field.

Answer: Thank you for your recommendation.